

# Pathways of topological rank analysis (PoTRA): a novel method to detect pathways involved in hepatocellular carcinoma

Chaoxing Li[1], Li Liu[2] and Valentin Dinu[2]

[1] School of Life Sciences, Arizona State University, Tempe, AZ, United States of America
[2] Department of Biomedical Informatics, Arizona State University, Scottsdale, AZ, United States of America

Corresponding author
Chaoxing Li, chaoxing@asu.edu

## ABSTRACT

Complex diseases such as cancer are usually the result of a combination of environmental factors and one or several biological pathways consisting of sets of genes. Each biological pathway exerts its function by delivering signaling through the gene network. Theoretically, a pathway is supposed to have a robust topological structure under normal physiological conditions. However, the pathway's topological structure could be altered under some pathological condition. It is well known that a normal biological network includes a small number of well-connected hub nodes and a large number of nodes that are non-hubs. In addition, it is reported that the loss of connectivity is a common topological trait of cancer networks, which is an assumption of our method. Hence, from normal to cancer, the process of the network losing connectivity might be the process of disrupting the structure of the network, namely, the number of hub genes might be altered in cancer compared to that in normal or the distribution of topological ranks of genes might be altered. Based on this, we propose a new PageRank-based method called Pathways of Topological Rank Analysis (PoTRA) to detect pathways involved in cancer. We use PageRank to measure the relative topological ranks of genes in each biological pathway, then select hub genes for each pathway, and use Fisher's exact test to test if the number of hub genes in each pathway is altered from normal to cancer. Alternatively, if the distribution of topological ranks of gene in a pathway is altered between normal and cancer, this pathway might also be involved in cancer. Hence, we use the Kolmogorov–Smirnov test to detect pathways that have an altered distribution of topological ranks of genes between two phenotypes. We apply PoTRA to study hepatocellular carcinoma (HCC) and several subtypes of HCC. Very interestingly, we discover that all significant pathways in HCC are cancer-associated generally, while several significant pathways in subtypes of HCC are HCC subtype-associated specifically. In conclusion, PoTRA is a new approach to explore and discover pathways involved in cancer. PoTRA can be used as a complement to other existing methods to broaden our understanding of the biological mechanisms behind cancer at the system-level.

## INTRODUCTION

High throughput technologies, such as genomic sequencing and microarrays, allow the genome-wide analysis of molecular factors associated with disease. While the technologies have advanced and have been refined to generate an increasing amount of high quality data, challenges remain in understanding the biological processes involved in disease etiology, particularly for complex disorders.

As we know, individual genomic alterations may result in diseases. For example, Hemophilia A is an X-linked recessive bleeding disorder caused by a deficiency in the activity of coagulation factor VIII (*Franchini & Mannucci, 2012*). Huntington's disease (HD) is an autosomal dominant progressive neurodegenerative disorder with a distinct phenotype characterized by chorea, dystonia, incoordination, cognitive decline, and behavioral difficulties, which is caused by a heterozygous expanded trinucleotide repeat (CAG)n, encoding glutamine, in the gene encoding huntingtin (HTT) on chromosome 4p16 (*Walker, 2007*; *Dayalu & Albin, 2015*).

In addition to monogenic diseases, many diseases are complex, such as diabetes, schizophrenia, or cancer, and are believed to be caused by a combination of genomic alterations, epigenetic, environmental and lifestyle factors (*Schork, 1997*; *Hindorff, Gillanders & Manolio, 2011*). Genomic disease association analysis suggests that complex diseases are not caused by individual genomic alterations. First, the complex disease phenotype is associated with many genes. Second, it may be associated with interactions among many genes. Therefore, more and more literature has been focusing on analyzing sets of genes associated with some phenotype. Gene expression profiles have been used to assess the activity of biological networks. Several approaches have been developed to identify active subnetworks across different phenotypes from changes in gene expression. One of the first such studies is a general approach to searching for ''active sub-networks'' associated with high levels of differential expression (*Ideker et al., 2002*). This approach identifies a set of genes that form a subnetwork whose expression is altered across two different phenotypes. Another very well-known method, Gene Set Enrichment Analysis (GSEA) (*Subramanian et al., 2005*), is a computational method that determines whether a pre-defined set of genes shows statistically significant, concordant differences between two phenotypes, which is also based on differential expression of a set of genes between two phenotypes. These approaches, while powerful and popular, are limited by the fact that they ignore the topology of the gene networks and sets that they investigate. Indeed, differential gene expression level in biological network might influence phenotypes. However, only investigating the differential expression levels of gene may be not sufficient since the topology of biological pathway is also an important characteristic of biological pathways and the role they play in both normal and pathological processes, as described below.

It is well known that the topological structure is very important for biological networks and it determines how genes interact with each other, governing how specific genes and biological pathways operate in the promotion or inhibition of human diseases (*Tavazoie et al., 1999*; *Goeman & Bühlmann, 2007*; *Tarca et al., 2009*; *Taylor et al., 2009*; *Khatri, Sirota & Butte, 2012*; *Rhinn et al., 2013*; *Mitrea et al., 2013*). Related to this, a hub gene within a

biological network is an important gene which acts to influence the activity of a number of genes (*Flintoft, 2004*), even influence the activity and function of the entire biological network. Hence, there has been an increased interest to analyze the co-regulation and co-expression of genes in the biological network, and many approaches have been developed to identify differential co-regulation and co-expression of genes in the subnetwork (*Kostka & Spang, 2004*; *Lai et al., 2004*; *Reverter et al., 2006*; *Watson, 2006*; *Choi & Kendziorski, 2009*; *Leonardson et al., 2010*; *Langfelder et al., 2011*; *Odibat & Reddy, 2012*). It has been a trend to extend differential expression analysis to differential network analysis (*De la Fuente, 2010*).

Most of the approaches for topology-based network and pathway analysis are based on different correlation-based metrics to identify differential networks between two different phenotypes. Generally, there are three main ways to compare networks for differential network analysis. The first approach handles weighted networks and uses some functions of the edge-specific weight differences as edge weights to construct differential networks (*Hudson, Reverter & Dalrymple, 2009*; *Tesson, Breitling & Jansen, 2010*; *Liu et al., 2010*; *Rhinn et al., 2013*). The second approach tries to find co-expressed gene sets and identify which correlation patterns are different between sets across conditions (*Watson, 2006*; *Rahmatallah, Emmert-Streib & Glazko, 2014*). This approach formulates summary measures that represent co-expression in a biological network and compares the metric between sets. The third approach compares the topology of biological networks across different phenotypes by using measures such as degree of nodes or modularity (*Reverter et al., 2006*; *Zhang et al., 2009*). However, the PoTRA method uses a topology-based metric to identify differential networks between two phenotypes. In addition to using different metrics, some of the other tools are based on correlation pattern of genes and identify groups of genes whose correlation patterns behave differentially across different datasets (*Watson, 2006*; *Hudson, Reverter & Dalrymple, 2009*; *Tesson, Breitling & Jansen, 2010*; *Liu et al., 2010*; *Rhinn et al., 2013*). Compared to these tools, PoTRA is directly based on topological ranks of genes and aims to identify pathways where the topological ranks of genes are different across datasets, which is more biologically intuitive. In this method, not only do we use correlation networks but we also use combined networks by taking intersected networks of correlation networks and KEGG curated pathways. Hence, when KEGG curated pathway information is employed, the topological rank-based PoTRA method can apply to the combined networks, while the previous correlation-based methods cannot, which is a limitation of the previously discussed correlation-only based methods. Regarding the previous tools based on topology (*Reverter et al., 2006*; *Zhang et al., 2009*), Zhang et al. focuses on identifying genes involved in topological changes, while PoTRA focuses on identifying pathways involved in topological changes. Also, Reverter et al. focuses on identifying genes with differential connectivity between two phenotypes, which is also different from PoTRA's application scenario.

Although the above methods for differential network analysis can deal with some important biological questions, they are still limited. In general, they are based on a basic hypothesis that some connections between genes across the groups could be thought of as ''passenger'' events and other connections are unique to either one of groups and thus could be ''driver'' events that contribute to disease progression and development. Hence,

they focus on the contribution of individual differential connections to disease. This results in several limitations. First, each differential connection is regarded by these methods to have an equal contribution to disease. However, it is well understood that loss of a connection between two hub genes from normal to disease is more deleterious than loss of a connection between two non-hub genes. Second, how differential connections ("driver" connections mentioned above) between pairs of genes are associated with diseases is still not very biologically intuitive, because how the dependency between genes contributes to diseases is usually little understood.

To address these problems, we propose a new PageRank-based method called Pathways of Topological Rank Analysis (PoTRA) to detect pathways associated with cancer. PageRank is an algorithm initially used by Google Search to rank websites in their search engine results (*Page et al., 1999*). It is a way of measuring the importance of nodes in a network. More generally, PageRank has been applied to other networks, e.g., social networks (*Pedroche et al., 2013*; *Wang et al., 2013*). To date, there have been several studies using PageRank for gene expression and network analysis (*Morrison et al., 2005*; *Winter et al., 2012*; *Kimmel & Visweswaran, 2013*; *Hou & Ma, 2014*; *Bourdakou, Athanasiadis & Spyrou, 2016*; *Zeng et al., 2016*; *Ramsahai et al., 2017*; *Morshed Osmani & Rahman, 2007*). These studies focus on ranking genes and discovering key driver genes in disease, and do not try to detect dysregulated pathways involved in disease. Other studies (*Winter et al., 2012*; *Zeng et al., 2016*) use PageRank to select topological important genes and simply see which pathways that these topological important genes are involved in. These PageRank-related approaches are very different from our approach.

Our approach embodied by PoTRA is motivated by the observation that the loss of connectivity is a common topological trait of cancer networks (*Anglani et al., 2014*), as well as the prior knowledge that a normal biological network includes a small number of well-connected hub nodes and a large number of nodes that are non-hubs (*Albert, 2005*; *Khanin & Wit, 2006*; *Zhu, Gerstein & Snyder, 2007*). However, from normal to cancer, the process of the network losing connectivity might be the process of disrupting the structure of the network, namely, the number of hub genes might be altered in cancer compared to that in normal or the distribution of topological ranks of genes might be altered. Thus, we hypothesize that if the number of hub genes is different in a pathway between normal and cancer, this pathway might be involved in cancer. Based on this hypothesis, we propose to detect pathways involved in cancer by testing if the number of hub genes for each pathway is different between normal and cancer samples.

Our approach embodied by PoTRA is also motivated by that the topological ranks of genes within biological pathways might be altered in cancer. Based on this, we propose to detect pathways involved in cancer by testing if the distribution of PageRank scores of genes for each pathway is altered between normal and cancer samples.

Therefore, the PoTRA approach computes topological ranks of genes in each pathway, and then detects pathways with significantly altered number of hub genes between normal and cancer, and, alternatively, detect pathways with significantly altered distributions of topological ranks of genes in corresponding pathways between two phenotypes. Namely, we first use the Google search PageRank algorithm to measure the relative topological ranks

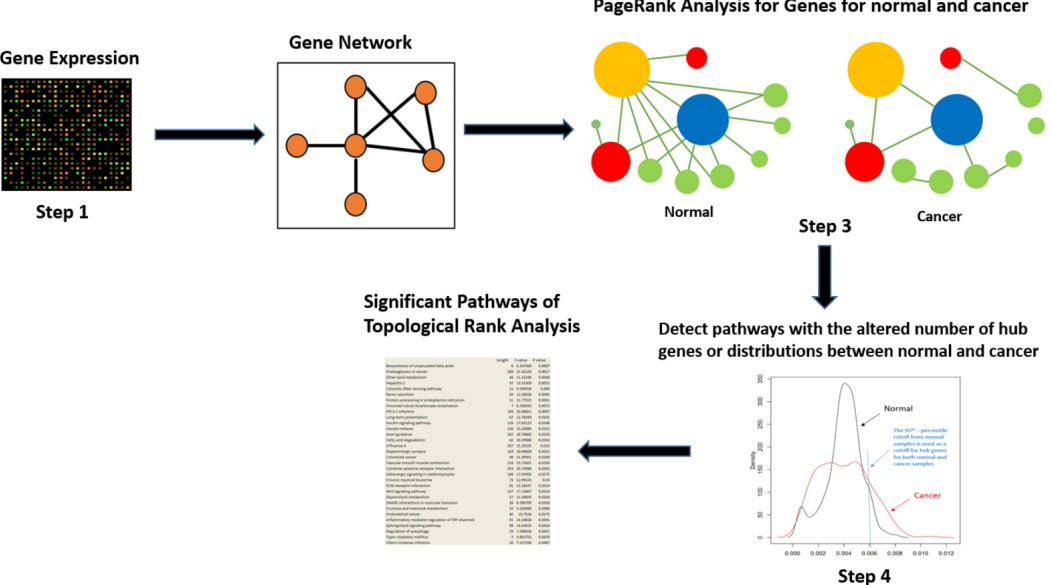

**Figure 1** Overview of the PoTRA method.

of genes in a biological pathway across different conditions. Then, we use Fisher's exact test to estimate if the number of hub genes in each pathway is significantly different between normal and cancer, and, alternatively, use the Kolmogorov–Smirnov test to estimate if the distribution of PageRank scores of genes in each pathway is significantly different between normal and cancer. As an illustration, we apply PoTRA to study hepatocellular carcinoma (HCC) and its subtypes and identify disease-relevant pathways. In conclusion, PoTRA is a new approach to explore and discover cancer-associated pathways. PoTRA can be used as a complement to other existing methods to enrich our understanding of the biological mechanisms behind cancer at the systems-level.

## MATERIALS AND METHODS
### Overview of the PoTRA method
Below we detail the steps of the PoTRA method, as illustrated in Fig. 1.

1. Data

   To illustrate the PoTRA method, we use publicly available gene expression datasets from The Cancer Genome Atlas (TCGA) (https://cancergenome.nih.gov/) hepatocellular carcinoma (HCC) study. We analyze and contrast 50 HCC samples and 50 tumor-adjacent normal samples ("normal samples" in future sections). In addition, the datasets also include gene expression profiles for several HCC subtypes. We further analyze and contrast 22 hepatitis B-induced HCC samples and 22 tumor-adjacent normal samples, 34 hepatitis C-induced HCC samples and 34 tumor-adjacent normal samples, and 50 alcohol-induced HCC samples and 50 tumor-adjacent normal samples. There are 20,531 gene expression values for each sample. Pathway information from the Kyoto Encyclopedia of Genes and

Genomes (KEGG) database (*Kanehisa & Goto, 2000*) is used. To date, there is much known about etiology of HCC (*Beasley, 1988*; *Sanyal, Yoon & Lencioni, 2010*; *Wang et al., 2012*; *Goossens & Hoshida, 2015*) and knowledge of pathways involved in HCC (*Villanueva et al., 2008*; *Zhou et al., 2010*; *Wang et al., 2017*), which makes it easier to illustrate and assess the PoTRA method.

2. Construction of gene co-expression network for a pathway

We apply the PoTRA method to gene expression profiles for several phenotypes, such as normal and cancer and cancer subtypes. First, we select genes for each pathway, using pathway information from KEGG. For each pathway, we determine the gene-gene interactions by using the Pearson's correlation to test each co-expressed gene pair. The test calculates a *P*-value for the association between each pair of genes. A significance threshold of 0.05 is used. When the *P*-value of a pair of genes is below 0.05, we establish an edge between the corresponding two genes; otherwise, there is no edge between them. We implement it through a built-in function called ''cor.test()'' in the statistical software package R (*R Core Team, 2013*). In this way, we can construct gene co-expression networks (i.e., pathways) for normal and cancer, separately. Of note, all the gene co-expression networks (i.e., pathways) used by PoTRA are undirected graphs, because co-expression networks only focus on gene pairs with a similar expression pattern across samples, in other words, the transcript levels of two co-expressed genes rise and fall together across samples.

In addition to construction of gene networks based on correlation alone, we, alternatively, also construct gene networks by combining the correlation with the pre-defined interaction from pathway databases.

3. PageRank analysis for genes within a pathway for normal and cancer

Based on the above constructed interactions within a pathway, we can observe that some genes are hub genes whereas others are non-hub genes. We apply the PageRank algorithm (*Page et al., 1999*) to obtain the corresponding topological importance for each gene within the pathway for normal and cancer, separately, see Fig. 2.

We implement it by using the page.rank() function from the igraph (*Csárdi & Nepusz, 2006*) R package. As mentioned in **Step 2**, all the networks that we construct are undirected graphs. Thus, the PageRank algorithm used in our approach is based on undirected graphs.

**The PageRank algorithm**

The PageRank algorithm is used by the Google search engine to rank the importance of web pages, which is based on the assumption that the importance of a web page is high in a network if this web page has connections with other nodes of high importance. This idea is naturally applied to analyzing biological networks, where the importance of a gene is high if this gene is connected to other genes of high importance. In our study, the gene-gene network is an undirected graph where a node represents a gene and the edges can be defined by prior knowledge (e.g., KEGG database).

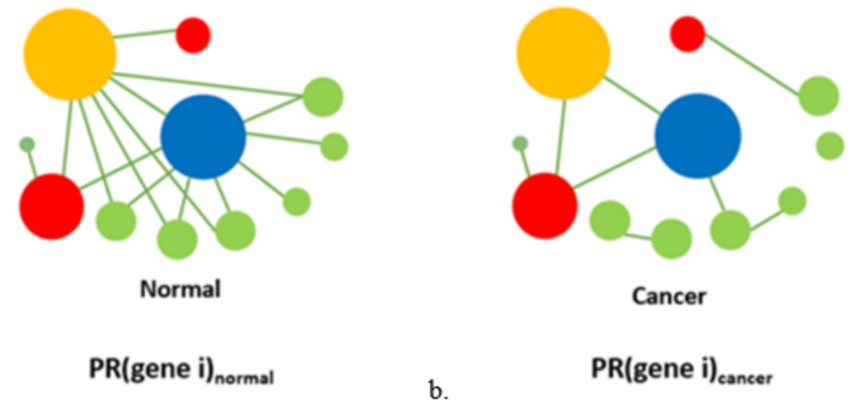

**Figure 2** **The topological rank analysis for each gene within a pathway.** For genes within a specified pathway, according to **Step 2**, we construct a corresponding gene co-expression network for normal and cancer, separately. Then we apply the PageRank method to obtain the topological importance of each gene for normal and cancer, separately. PR(gene i)$_{normal}$ represents the PageRank score of the gene i for normal samples, while PR(gene i)$_{cancer}$ represents the PageRank score of the gene i for cancer samples.

The output from the PageRank algorithm is a probability distribution representing the likelihood that a person randomly clicking on links will arrive at any particular web page. A probability is a numeric value between 0 and 1. The sum of probabilities for all web pages is equal to 1. The probability of a web page is proportional with the time spent at the web page when a person surfs the web. This idea can also be intuitively extended to ranking genes in gene networks where the probability of a gene is proportional with the time a research scientist spends looking and returning at the same gene when analyzing research results. For additional details of PageRank, please refer to *Page et al. (1999)*.

4. Detect pathways with significant changes between normal and cancer scores

4. 1 Detect pathways with significantly altered number of hub genes between normal and cancer using Fisher's exact test

As mentioned above, PoTRA is motivated by the observation that the loss of connectivity is a common topological trait of cancer networks (*Anglani et al., 2014*) and the prior knowledge that a normal biological network includes a small number of well-connected hub nodes and a large number of nodes that are non-hubs (*Albert, 2005*; *Khanin & Wit, 2006*; *Zhu, Gerstein & Snyder, 2007*). From normal to cancer, the process of the network losing connectivity might be the process of disrupting the structure of the network, which can result in an altered number of hub genes between normal and cancer. Hence, a statistics that we compare between two phenotypes is the number of hub genes. The PageRank scores of all genes of a pathway form a distribution, and we use the 95th percentile of the distribution (one-tail) in normal samples as cutoff value for hub genes for both normal and cancer samples. The genes in this pathway with PageRank scores that are above the cutoff value are identified as hub genes for this pathway. Then we count the number of hub genes for normal and cancer, separately. Next, we use Fisher's exact test to assess if the

**Table 1  Contingency table for Fisher's exact test.** We use the 95th percentile of the distribution (one-tail) in normal samples as cutoff value for hub genes for both normal and cancer samples. The value "a" represents the number of genes whose PageRank scores are below the cutoff value for normal samples. The value "b" represents the number of genes whose PageRank scores are above the cutoff value for normal samples. The values "c" and "d" are the corresponding values for cancer. We use Fisher's exact test to assess if the number of hub genes is significantly different between normal and cancer.

|  | Number of non-hub genes | Number of hub genes | Total |
|---|---|---|---|
| Normal | a | b | a+b |
| Cancer | c | d | c+d |
| Total | a+c | b+d | a+b+c+d |

number of hub genes is significantly different between normal and cancer. For details, see Table 1.

Fisher's exact test estimates the probability of obtaining any such set of values, given by the hypergeometric distribution:

$$P = \frac{\binom{a+b}{a}\binom{c+d}{c}}{\binom{n}{a+c}} = \frac{(a+b)!(c+d)!(a+c)!(b+d)!}{a!b!c!n!} \tag{1}$$

where $n = a+b+c+d$, and $\binom{i}{j}$ is the binomial coefficient and the symbol "!" indicates the factorial operator.

**Formula 1** gives the exact hypergeometric probability of observing this particular arrangement of the data, assuming the given marginal totals, on the null hypothesis that the number of hub genes is the same for a specified pathway between normal and cancer. If this test statistic is significant, it indicates that there is a significantly different number of hub genes between normal and cancer, thereby this pathway might be involved in cancer. By studying many pathways from the KEGG database we generate a multiple hypothesis testing problem. We address this issue by correcting the $P$-values calculated for each pathway using the False Discovery Rate (FDR) approach, using the Benjamini and Hochberg procedure (*Benjamini & Hochberg, 1995*).

4.2. Detect pathways with significantly altered distributions of PageRank scores for genes between normal and cancer using Kolmogorov–Smirnov test

Alternatively, PoTRA is implemented by testing if the distribution of PageRank scores of genes is altered between normal and cancer, using the two-sample Kolmogorov–Smirnov test. The two-sample Kolmogorov–Smirnov test is a nonparametric test of equality of continuous, one-dimensional probability distributions that can be used to compare two samples. The Kolmogorov–Smirnov statistic quantifies a distance between the empirical distribution functions of two samples. The null distribution of this statistic is calculated under the null hypothesis that the samples are drawn from the same distribution.

The empirical distribution function $F_n$ for $n$ i.i.d. observation $X_i$ is defined as:

$$F_n(x) = \frac{1}{n}\sum_{i=1}^{n} I_{|-\infty,x|}(X_i) \tag{2}$$

where $I_{|-\infty,x|}(X_i)$ is the indicator function, equal to 1 if $X_i \leq x$ and equal to 0 otherwise.

The two-sample Kolmogorov–Smirnov test may be used to test if two underlying one-dimensional probability distributions differ. In this case, the Kolmogorov–Smirnov statistic is:

$$D_{n,m} = \sup_x |F_{1,n}(x) - F_{2,m}(x)| \tag{3}$$

where $F_{1,n}$ and $F_{2,m}$ are the empirical distribution functions of the first and the second sample respectively, and sup is the supremum function.

If the test statistic for a specific pathway is significant, it indicates that the two underlying distributions differ between normal and cancer, thereby this pathway is involved in cancer. By studying many pathways from the KEGG database we generate a multiple hypothesis testing problem. We address this issue by correcting the $P$-values calculated for each pathway using the False Discovery Rate (FDR) approach, using the Benjamini and Hochberg procedure (*Benjamini & Hochberg, 1995*).

**Software tools**

All the analysis is conducted using the R statistical programming language (*R Core Team, 2013*), using the following R Bioconductor packages: graphite for pathway databases (*Sales, Calura & Romualdi, 2017*), igraph for PageRank function (*Csárdi & Nepusz, 2006*) and graph for visualization (*Gentleman et al., 2017*).

# RESULTS

We apply PoTRA to analyze and contrast 50 HCC samples and 50 tumor-adjacent normal samples. All data come from The Cancer Genome Atlas (TCGA) hepatocellular carcinoma (HCC) study.

## PoTRA for HCC vs. normal samples using correlation-based networks

To illustrate the PoTRA method, we use a cancer-associated pathway, the "MAPK signaling pathway", as an example in the following section.

### Construction of a gene co-expression network for a pathway

As suggested before, the "MAPK signaling pathway" might be comprised of different interactions between genes under different conditions, such as normal versus cancer conditions. First, we need to find the genes that this pathway consists of by using the KEGG database. In practice, we implement it by using an R package called graphite. Second, for the genes of this pathway, we identify the interactions between genes for normal and cancer samples, separately. There are 14,005 edges for normal and 5,170 edges for cancer. For the information of the "MAPK signaling pathway" in normal and cancer samples, see Table S1.

### PageRank analysis for genes within a pathway for normal and cancer

Based on the interactions identified in the previous section, we can obtain a PageRank score for each gene in the "MAPK signaling pathway" for normal and cancer, separately, which quantifies the influence of a gene on the activity of other genes in this pathway. For the results for this step, two vectors with 250 PageRank values, one for normal and one
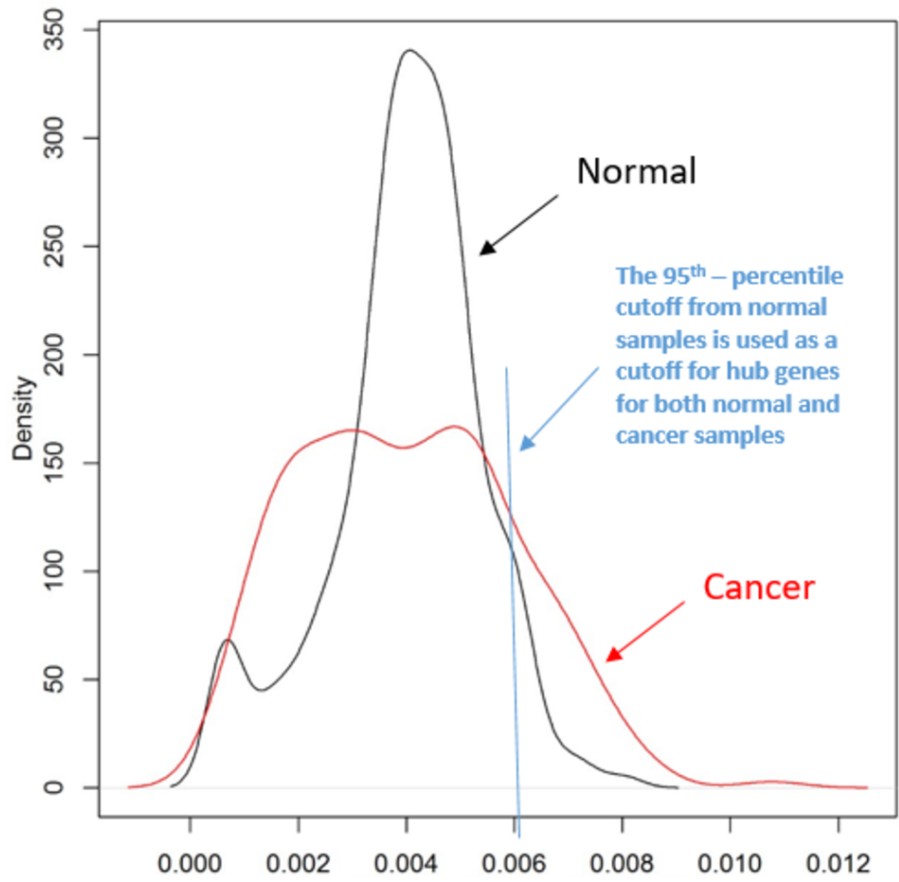

**Figure 3  The kernel density distribution of PageRank scores of genes in "MAPK signaling pathway".**
The red line shows the kernel density distribution of PageRank scores for cancer and the black one is for
normal samples. Note that the mean for the two distributions is the same, i.e., mean $= 1/N = 0.004$, where
$N = 250$ is the number of genes in the "MAPK signaling pathway" pathway. We use the 95th-percentile
cutoff ($=0.006035$) of the kernel distribution in normal samples as cutoff for hub genes for both normal
and cancer samples.

for cancer, separately, see Table S1. As previously mentioned, the PageRank values in each
vector add up to 1.

Figure 3 illustrates the distributions of PageRank scores for genes in "MAPK signaling
pathway" for normal and cancer, separately.

The number of hub genes changes. While there are only 13, strongly-connected (with
more edges, an average of 121 edges) hub genes in the normal samples, there are 40 hub
genes in the cancer samples, more loosely-connected (with fewer edges, an average of 75
edges).

As mentioned before, the process of the network losing connectivity might be the process
of disrupting the structure of the network which includes a small number of hub nodes
and a large number of non-hub nodes, namely, the number of hub genes might be altered
in cancer compared to that in normal. The altered number of hub genes might be a trait of

**Table 2** **The "MAPK signaling pathway" pathway identified by PoTRA for HCC using Fisher's exact test.** The *P* value is adjusted by False Discovery Rate (FDR).

| | Gene Count(L) | # of edges_normal | # of edges_cancer | # of hub genes_normal | # of hub genes_cancer | Adjusted *P*-value |
|---|---|---|---|---|---|---|
| MAPK signaling pathway | 250 | 14,005 | 5,170 | 13 | 40 | 0.0158 |

**Table 3** **The significant KEGG pathways identified by PoTRA for HCC using Fisher's exact test.** FDR adjusted *P*-values are below 0.05.

| | | Gene Count(L) | # of edges_normal | # of edges_cancer | # of hub genes_normal | # of hub genes_cancer | Adjusted *P*-value |
|---|---|---|---|---|---|---|---|
| 1 | Pathways in cancer | 310 | 24,924 | 9,136 | 16 | 48 | 0.0081 |
| 2 | MAPK signaling pathway | 252 | 14,005 | 5,170 | 13 | 40 | 0.0158 |
| 3 | Breast cancer | 143 | 3,589 | 1,175 | 8 | 29 | 0.0278 |

a pathway in cancer, which suggests the pathway is involved in cancer if the change in hub gene number is statistically significant.

Importantly, we find increased variance (median change 5.2 -fold for the MAPK pathway, or 6.1 fold change for all genes) of gene expression in cancer samples compared to normal samples, which is included in Table S1. The increased variance in cancer samples results in lower Pearson correlations between genes in cancer, which leads to lower correlations and loss of connectivity in cancer. This loss of connectivity can lead to the disruption of the structure of the biological networks. To quantify these changes in the network topology between normal and disease, we use Fisher's exact test to identify changes in the number of network hub genes, and the Kolmogorov Smirnov test to test if the distributions of PageRank scores are significantly different between normal and cancer. These tests are described below.

*Fisher's exact test for comparing the number of hub genes in the pathway.* We next use Fisher's exact test to test if the number of hub genes for the "MAPK signaling pathway" is significantly different between normal and cancer. The result for the "MAPK signaling pathway" pathway is included in Table 2.

The low *P*-value in Table 2 indicates that the number of hub genes in cancer samples is significantly different from that in normal samples, suggesting that the "MAPK signaling pathway" pathway is involved in HCC. This example suggests that a normal biological network includes a small number of hub nodes and a large number of non-hub nodes. Moreover, the loss of connectivity from normal to cancer might lead to disrupting the structure of the network in cancer, thereby resulting in the fact that the number of hub genes is altered in cancer compared to that in normal.

Then we apply the same approach to other pathways from KEGG to compare HCC vs. normal samples. The significant pathways are shown in Table 3.

We find three significant pathways with altered number of hub genes between normal and cancer. It is well known that these three pathways are strongly associated with cancer in general. The MAPK signaling pathway plays a role in the regulation of gene expression, cellular growth, and survival (*Knight & Irving, 2014*). Abnormal MAPK signaling might

**Table 4  The "MAPK signaling pathway" pathway identified by PoTRA for HCC using the Kolmogorov–Smirnov test.** The *P* value is adjusted by False Discovery Rate (FDR).

|  | Gene Count(L) | # of edges_normal | # of edges_cancer | Adjusted *P*-value |
|---|---|---|---|---|
| MAPK signaling pathway | 250 | 14,005 | 5,170 | 0.0278423 |

**Table 5  The significant KEGG pathways identified by PoTRA for HCC using the Kolmogorov–Smirnov test.** FDR adjusted *P*-values are below 0.05.

|  |  | Gene Count(L) | # of edges_normal | # of edges_cancer | Adjusted *P*-value |
|---|---|---|---|---|---|
| 1 | RNA transport | 133 | 7,168 | 5,343 | 6.72E−08 |
| 2 | mRNA surveillance pathway | 70 | 1,867 | 1,252 | 0.023328272 |
| 3 | MAPK signaling pathway | 250 | 14,005 | 5,170 | 0.027842298 |

lead to uncontrolled or increased cell proliferation and resistance to apoptosis (*Santarpia, Lippman & El-Naggar, 2012*; *Burotto et al., 2014*). Interestingly, we also find that loss of connectivity and the larger number of hub genes for cancer are characteristics of the other two pathways as well.

The levels for parameter thresholds can be varied. Hence, we also have tried different threshold levels for hub genes in Fisher's exact test: 0.9, 0.85, 0.8, 0.75 and 0.7. For these threshold levels, the *p*-values are larger than that when the threshold is 0.95, and we do not find any significant pathways using Fisher's exact test. This suggests that, for this test, using a limited number of genes of higher "hubness" (higher PageRank values) is important when comparing normal and cancer.

*The Kolmogorov–Smirnov test for comparing distributions of PageRank scores of genes for pathways.*  We next use the Kolmogorov–Smirnov test to test if the two distributions of PageRank scores of genes for the "MAPK signaling pathway" are significantly different between normal and cancer. The result for the "MAPK signaling pathway" pathway is included in Table 4.

The low *P*-value in Table 4 indicates that the distribution of PageRank scores of genes in cancer samples is significantly different from that in normal samples, suggesting that the "MAPK signaling pathway" pathway is involved in HCC. The MAPK signaling pathway plays a role in the regulation of gene expression, cellular growth, and survival (*Knight & Irving, 2014*). Abnormal MAPK signaling might lead to uncontrolled or increased cell proliferation and resistance to apoptosis (*Santarpia, Lippman & El-Naggar, 2012*; *Burotto et al., 2014*).

Then we apply the same approach to other pathways from KEGG to compare HCC with normal samples. The significant pathways are shown in Table 5.

We find three significant pathways with altered distribution of PageRank scores of genes between normal and cancer. In addition to the "MAPK signaling pathway", we find two other pathways: "RNA transport" and the "mRNA surveillance pathway". For the "RNA transport" pathway, it is well known that the nuclear export of mRNA is intrinsically

**Table 6  The significant KEGG pathways identified by PoTRA for hepatitis B-induced HCC using Fisher's exact test.** FDR adjusted *P*-values are below 0.05.

|  |  | Gene Count(L) | # of edges_normal | # of edges_cancer | # of hub genes_normal | # of hub genes_cancer | Adjusted *P*-value |
|---|---|---|---|---|---|---|---|
| 1 | Insulin signaling pathway | 139 | 2,692 | 958 | 7 | 34 | 0.0007 |
| 2 | Pathways in cancer | 310 | 11,194 | 3,792 | 16 | 52 | 0.0007 |
| 3 | Hippo signaling pathway | 151 | 2,836 | 970 | 8 | 31 | 0.0072 |
| 4 | HTLV-I infection | 194 | 5,518 | 2,080 | 10 | 35 | 0.0072 |
| 5 | Neurotrophin signaling pathway | 117 | 2,441 | 895 | 6 | 25 | 0.0195 |
| 6 | mTOR signaling pathway | 144 | 34,10 | 832 | 8 | 28 | 0.0240 |
| 7 | Epstein-Barr virus infection | 85 | 1,524 | 435 | 5 | 21 | 0.0353 |
| 8 | Hepatitis B | 134 | 2,708 | 828 | 7 | 25 | 0.0353 |

linked to the co-transcriptional processing of nascent transcripts synthesized by the RNA polymerase II. This functional coupling is very important for the survival of cells allowing for nuclear export of fully processed transcripts, which could lead to genome instability and to various forms of cancer (*Siddiqui & Borden, 2012*; *Hautbergue, 2017*). The "mRNA surveillance pathway" is a quality control mechanism that detects and degrades abnormal mRNAs, which includes nonsense-mediated mRNA decay (NMD), nonstop mRNA decay (NSD), and no-go decay (NGD). NMD can degrade PTC (Premature termination codons)-containing transcripts which cause a large proportion of human cancers (*Lindeboom, Supek & Lehner, 2016*). Nonstop decay is the mechanism of identifying and disposing aberrant transcripts that lack in-frame stop codons. These transcripts are identified during translation when the ribosome arrives at the 3′ end of the mRNA and stalls at the end of the poly(A) tail. This surveillance mechanism protects the cells from potentially harmful truncated proteins, but it may also be involved in mediating critical cellular functions of transcripts that are prone to stop codon read-through, which have implications in human cancers (*Klauer & Van Hoof, 2012*). No-go decay (NGD) is a eukaryotic quality control mechanism that evolved to cope with translational arrests (*Simms, Yan & Zaher, 2017*). These above processes are strongly related to cancer.

### PoTRA for cancer subtype analysis

*Fisher's exact test for cancer subtype analysis.* Many complex diseases have subtypes and/or can be classified into different categories based on diagnosis, pathology, phenotype characteristics, etc. To further assess the PoTRA method, we apply it to several subtypes of the HCC TCGA data. There are several risk factors associated with HCC, such as hepatitis B, hepatitis C and alcohol (*Beasley, 1988*; *Sanyal, Yoon & Lencioni, 2010*; *Hoshida et al., 2014*; *Goossens & Hoshida, 2015*). Here, we apply PoTRA to compare these three subtypes of HCC samples with normal samples.

Table 6 illustrates the Fisher's exact test results for comparing normal with hepatitis B-induced HCC samples.

There is one common pathway, Pathways in cancer, between Tables 6 and 3. There are seven other new pathways, which are very interesting and associated with the hepatitis B-induced HCC. First, the "Hepatitis B" pathway is detected by our method. Hepatitis

**Table 7  The significant KEGG pathways identified by PoTRA for hepatitis C-induced HCC using Fisher's exact test.** FDR adjusted *P*-values are below 0.05.

|   |   | Gene Count(L) | # of edges_normal | # of edges_cancer | # of hub genes_normal | # of hub genes_cancer | Adjusted *P*-value |
|---|---|---|---|---|---|---|---|
| 1 | Pathways in cancer | 310 | 22,253 | 7,791 | 16 | 62 | 2.89E−06 |
| 2 | PI3K-Akt signaling pathway | 340 | 19,901 | 6,594 | 17 | 65 | 2.89E−06 |
| 3 | MAPK signaling pathway | 252 | 11,986 | 4,168 | 13 | 47 | 0.0003 |
| 4 | Proteoglycans in cancer | 204 | 9,815 | 3,642 | 11 | 38 | 0.0033 |
| 5 | Rap1 signaling pathway | 208 | 8,294 | 3,587 | 11 | 34 | 0.0215 |
| 6 | Adrenergic signaling in cardiomyocytes | 149 | 3,594 | 1,355 | 8 | 27 | 0.0372 |
| 7 | cAMP signaling pathway | 196 | 5,106 | 2,493 | 10 | 30 | 0.0372 |
| 8 | Focal adhesion | 203 | 10,225 | 4,656 | 11 | 32 | 0.0372 |
| 9 | HTLV-I infection | 194 | 9,843 | 4,030 | 10 | 30 | 0.0372 |
| 10 | Ras signaling pathway | 226 | 10,098 | 3,931 | 12 | 33 | 0.0376 |
| 11 | FoxO signaling pathway | 126 | 3,391 | 1,222 | 7 | 24 | 0.0380 |
| 12 | Osteoclast differentiation | 123 | 4,418 | 1,452 | 7 | 24 | 0.0380 |
| 13 | ErbB signaling pathway | 88 | 2,128 | 814 | 5 | 20 | 0.0400 |
| 14 | Axon guidance | 167 | 6,203 | 2,705 | 9 | 27 | 0.0433 |

B is the most important and direct factor causing hepatitis B-induced HCC. In addition, we find two other pathways, HTLV-I (Human T-cell leukemia virus type I) infection and Epstein-Barr virus infection, which are strongly associated with virus infection and cancer. This is consistent with the viral pathology of hepatitis B-induced HCC. Besides, some studies show that hepatitis B virus infection can contribute to the impairment of insulin signaling, which is another pathway identified by PoTRA (*Kim, Kim & Cheong, 2010*; *Barthel et al., 2016*). Finally, the other three pathways, Hippo signaling pathway, Neurotrophin signaling pathway and mTOR signaling pathway are associated with cancer in general. Hippo signaling pathway is reported to be able to control organ size through regulating cell proliferation and apoptosis (*Saucedo & Edgar, 2007*; *Pan, 2010*). It is reported that neurotrophins can regulate cancer stem cells (*Chopin et al., 2016*), and neurotrophins contribute to pro-survival signaling in many different types of cancer (*Molloy, Read & Gorman, 2011*). The mTOR signaling pathway is a well-known cancer-associated pathway. Alterations of mTOR signaling pathway have significant effects on cancer progression. The major components of mTOR signaling pathway are critical effectors in cell signaling pathways commonly deregulated in cancers (*Guertin & Sabatini, 2007*; *Villanueva et al., 2008*; *Pópulo, Lopes & Soares, 2012*).

These results suggest that PoTRA can be used to identify not only the pathways associated with cancer in general, but also those pathways associated with cancer subtypes, such as hepatitis B-induced HCC specifically.

Results of the PoTRA analysis from two other HCC subtypes, hepatitis C-induced HCC and alcohol-induced HCC, are included in Tables 7 and 8, respectively.

In Table 5, we find two common pathways, Pathways in cancer and the MAPK signaling pathway, between Tables 5 and 3. Among the other pathways, we find several pathways related to cancer generally and hepatitis C-induced HCC specifically. First, HTLV-I

**Table 8** **The significant KEGG pathways identified by PoTRA for alcohol-induced HCC using Fisher's exact test.** FDR adjusted *P*-values are below 0.05.

| | | Gene Count(L) | # of edges_normal | # of edges_cancer | # of hub genes_normal | # of hub genes_cancer | Adjusted *P*-value |
|---|---|---|---|---|---|---|---|
| 1 | PI3K-Akt signaling pathway | 340 | 23,928 | 8,733 | 17 | 55 | 0.0006 |
| 2 | MAPK signaling pathway | 252 | 14,005 | 5,767 | 13 | 46 | 0.0007 |
| 3 | Pathways in cancer | 310 | 24,924 | 10,191 | 16 | 47 | 0.0043 |

infection is also listed in this table, and, as mentioned above, is associated with virus infection and cancer. Almost all other pathways are associated with cancer in general. PI3K-Akt signaling pathway is a key regulator of normal cellular processes involved in cell growth, proliferation, motility, survival, and apoptosis (*Porta, Paglino & Mosca, 2014*). The Proteoglycans in cancer pathway is involved in regulation of proteoglycans, heavily glycosylated proteins present especially in connective tissue in cancer. The Rap1 signaling pathway is reported to be involved in cancer cell migration, invasion and metastasis (*Bailey, 2009*; *Zhang et al., 2017*). The cAMP signaling pathway regulates a number of biological processes, such as cell growth and adhesion, neuronal signaling, energy homeostasis and muscle relaxation (*Fajardo, Piazza & Tinsley, 2014*). The key component of Focal adhesion pathway, focal adhesion kinase (FAK), is reported to enable activation by growth factor receptors or integrins in different types of cancers. FAK is an important mediator of cell proliferation, cell migration, cell growth (*Golubovskaya, Kweh & Cance, 2009*; *Tai, Chen & Shen, 2015*). A large volume of literature shows that the Ras signaling pathway is involved in several aspects of normal cell growth and malignant transformation, and plays an important role in cancer development and progression (*Vojtek & Der, 1998*; *Downward, 2003*; *Santarpia, Lippman & El-Naggar, 2012*; *Knight & Irving, 2014*). The FoxO signaling pathway is involved in the regulation of the cell cycle, apoptosis and metabolism (*Schmidt et al., 2002*; *Fu & Tindall, 2008*; *Gross, Van den Heuvel & Birnbaum, 2008*). Besides, activity of the FoxO signaling pathway also affects stem cell maintenance and lifespan (*Eijkelenboom & Burgering, 2013*). ErbB signaling pathway plays roles in cancer development and progression (*Hynes & Lane, 2005*; *Seshacharyulu et al., 2012*), as well as in cancer cell migration and invasion (*Appert-Collin et al., 2015*). The ErbB signaling pathway is associated with the development of a wide variety of types of solid tumor if ErbB signaling is excessive (*Cho & Leahy, 2002*). The Axon guidance pathway is also reported to regulate cell migration and apoptosis, and be associated with tumorigenesis (*Chédotal, Kerjan & Moreau-Fauvarque, 2005*).

We find two common pathways between Tables 8 and 3, MAPK signaling pathway and Pathways in cancer. As mentioned above, PI3K-Akt signaling pathway also plays an important role in cancer (*Porta, Paglino & Mosca, 2014*).

*Kolmogorov–Smirnov test for cancer subtype analysis.* Table 9 illustrates the Kolmogorov–Smirnov test results for comparing normal with hepatitis C-induced HCC samples.

In Table 9, we find two common pathways, the RNA transport and the MAPK signaling pathway, between Tables 9 and 5. Among the other pathways, we find pathways related

**Table 9  The significant KEGG pathways identified by PoTRA for hepatitis C-induced HCC using the Kolmogorov–Smirnov test.** FDR adjusted *P*-values are below 0.05.

|   |   | Gene Count(L) | # of edges_normal | # of edges_cancer | Adjusted *P*-value |
|---|---|---|---|---|---|
| 1 | RNA transport | 133 | 6,877 | 4,001 | 3.33E−06 |
| 2 | Pathways in cancer | 310 | 22,253 | 7,791 | 0.0055638 |
| 3 | Proteoglycans in cancer | 204 | 9,815 | 3,642 | 0.0055638 |
| 4 | MAPK signaling pathway | 250 | 11,986 | 4,168 | 0.01535456 |
| 5 | PI3K-Akt signaling pathway | 340 | 19,901 | 6,594 | 0.01975723 |
| 6 | HTLV-I infection | 194 | 9,843 | 4,030 | 0.01975723 |

**Table 10  The significant KEGG pathways identified by PoTRA for alcohol-induced HCC using the Kolmogorov–Smirnov test.** FDR adjusted *P*-values are below 0.05.

|   |   | Gene Count(L) | # of edges_normal | # of edges_cancer | Adjusted *P*-value |
|---|---|---|---|---|---|
| 1 | RNA transport | 133 | 7,168 | 4,816 | 2.94E−08 |
| 2 | Pathways in cancer | 310 | 24,924 | 10,191 | 0.0062832 |
| 3 | PI3K-Akt signaling pathway | 340 | 23,928 | 8,733 | 0.0062832 |
| 4 | MAPK signaling pathway | 250 | 14,005 | 5,767 | 0.01050039 |
| 5 | mRNA surveillance pathway | 70 | 1,867 | 1,096 | 0.01166414 |

**Table 11  The significant KEGG pathways identified by PoTRA for hepatitis B-induced HCC using the Kolmogorov–Smirnov test.** FDR adjusted *P*-values are below 0.05.

|   |   | Gene Count(L) | # of edges_normal | # of edges_cancer | Adjusted *P*-value |
|---|---|---|---|---|---|
| 1 | Arginine and proline metabolism | 50 | 163 | 26 | 0.005412 |
| 2 | Glyoxylate and dicarboxylate metabolism | 26 | 216 | 19 | 0.02765948 |
| 3 | Primary bile acid biosynthesis | 17 | 62 | 6 | 0.02765948 |
| 4 | Insulin signaling pathway | 139 | 2,692 | 958 | 0.04057992 |
| 5 | Vasopressin-regulated water reabsorption | 22 | 58 | 8 | 0.04057992 |

to cancer generally and hepatitis C-induced HCC specifically. First, HTLV-I infection is also listed in this table, and, as mentioned above, is associated with virus infection and cancer. PI3K-Akt signaling pathway is a key regulator of normal cellular processes involved in cell growth, proliferation, motility, survival, and apoptosis (*Porta, Paglino & Mosca, 2014*). Importantly, hepatitis C virus activates PI3K-Akt signaling to enhance entry and replication, and meanwhile PI3K-Akt signaling pathway also can increase HCV translation (*Liu et al., 2012*; *Shi, Hoffman & Liu, 2016*), which suggest that PI3K-Akt signaling pathway is associated with hepatitis C virus infection specifically. The "Pathways in cancer" pathway is associated with cancer in general. The "Proteoglycans in cancer" pathway is involved in regulation of proteoglycans, heavily glycosylated proteins present especially in connective tissue in cancer (*Iozzo & Sanderson, 2011*; *Baghy et al., 2016*).

Results of the PoTRA analysis from two other HCC subtypes, alcohol-induced HCC and hepatitis B-induced HCC, are included in Tables 10 and 11, respectively.

We find three common pathways between Tables 10 and 5: RNA transport, the MAPK signaling pathway and the mRNA surveillance pathway. As mentioned above, the "Pathways

**Table 12  The significant KEGG pathways identified by PoTRA for hepatitis C-induced HCC using the Fisher's exact test based on combined networks.** FDR adjusted $P$-values are below 0.05. E.comb.normal represents the number of edges in the combined network for normal samples, while E.comb.case is for cancer samples, respectively.

|   |   | Gene Counts | E.comb.normal | E.comb.case | Adjusted $P$ value |
|---|---|---|---|---|---|
| 1 | Epstein-Barr virus infection | 85 | 131 | 43 | 0.0103488 |
| 2 | p53 signaling pathway | 68 | 57 | 20 | 0.0103488 |

in cancer" pathway is strongly associated with cancer in general and the PI3K-Akt signaling pathway also plays an important role in cancer (*Porta, Paglino & Mosca, 2014*).

There is no common pathway between Tables 11 and 5. Very interestingly, for the association between hepatitis B-induced HCC and the "Primary bile acid biosynthesis" pathway, some studies demonstrate that hepatitis B virus infection can alter bile acid metabolism as a consequence of impaired bile acid uptake (*Oehler et al., 2014*; *Geier, 2014*). Also, some studies show that hepatitis B virus infection can contribute to the impairment of insulin signaling (*Kim, Kim & Cheong, 2010*; *Barthel et al., 2016*). These two pathways mentioned are specifically related to hepatitis B virus infection. Arginine and proline metabolism is one of the central pathways for the biosynthesis of the amino acids arginine and proline from glutamate. Some studies have suggested that altered arginine and proline metabolism is linked to metastasis formation in cancer (*Elia et al., 2017*). Arginine serves as an intermediate in the urea cycle and as a precursor for protein, polyamine, creatine and nitric oxide (NO) biosynthesis. NO may influence tumor initiation, promotion, and progression, tumor-cell adhesion, apoptosis angiogenesis, differentiation, chemosensitivity, radiosensitivity, and tumor-induced immunosuppression (*Lind, 2004*).

## PoTRA for HCC vs. normal samples using networks that combine correlation-based networks with curated interaction networks

The above results show that the Fisher's exact test and the Kolmogorov–Smirnov test can identify cancer-related pathways based on correlation networks. In addition to correlation networks, in this section we apply PoTRA to gene networks constructed by intersecting correlation networks with pre-defined networks from the KEGG database. As performed in the previous section we investigate if Fisher's exact test and Kolmogorov–Smirnov test are still able to robustly discover cancer-associated pathways and differential pathways between HCC and subtypes of HCC based on the combined networks.

### Fisher's exact test for HCC and subtypes of HCC

For HCC, we use Fisher's exact test and identify no significant pathway with altered number of hub genes between normal and HCC.

For hepatitis C-induced HCC, we identify two significant pathways, listed in Table 12.

As mentioned above, the Epstein-Barr virus infection is a pathway associated with virus infection, which is specifically related to hepatitis C induced HCC. P53 signaling pathway is a classical oncogenic pathway, and it can regulate the cell cycle, apoptosis and help prevent cancer. The major component of p53 signaling pathway, p53 protein, is most frequently altered in human cancer (*May & May, 1999*; *Sherr & McCormick, 2002*; *Sui et al., 2011*; *Stegh, 2012*).

**Table 13 The significant KEGG pathways identified by PoTRA for hepatitis B-induced HCC using the Fisher's exact test based on combined networks.** FDR adjusted $P$-values are below 0.05. E.comb.normal represents the number of edges in the combined network for normal samples, while E.comb.case is for cancer samples, respectively.

| | | Gene Counts | E.comb.normal | E.comb.case | Adjusted $P$ value |
|---|---|---|---|---|---|
| 1 | Epstein-Barr virus infection | 85 | 84 | 22 | 0.0103488 |

For hepatitis B-induced HCC, we identify one significant pathway, listed in Table 13.

As such, we identify the same significant pathway "Epstein-Barr virus infection", which is also specific for hepatitis C-induced HCC, discussed above.

For alcohol-induced HCC, we identify no significant pathways.

First, we can observe a large loss of connectivity for combined networks vs. correlation networks in normal and in cancer. For example, Table 13 lists only 84 edges in normal and 22 edges in cancer samples for the Epstein-Barr virus infection pathway using combined correlation and curated interactions. This is in contrast to 1,524 edges in normal and 435 edges in cancer shown in Table 6 for the same pathway, when using all the correlation-based edges. We can observe that the proportion of edges in normal to cancer for the correlation network and the combined network are approximately the same (about 4:1). On average, we observe a reduction by, a factor of 23.18 in the number of edges between correlation and combined networks. However, the PoTRA-Fisher's exact test still can identify pathways specifically involved in subtypes of HCC.

Furthermore, because combined networks have a large loss of connectivity from correlation networks to combined networks, the PageRank scores of genes are more evenly distributed in combined networks than in correlation networks. Hence, fewer significant pathways with altered number of hub genes between normal and cancer are identified for combined networks than that for correlation networks. Although Fisher's exact test identifies fewer significant pathways, this test can still discover differential pathways specific for subtypes of HCC between normal and cancer, suggesting that the Fisher's exact test is able to robustly discover pathways involved in cancer and subtypes of cancer.

### Kolmogorov–Smirnov test for HCC and subtypes of HCC

All significant pathways for HCC and each subtype of HCC (hepatitis C-induced HCC, hepatitis B-induced HCC and alcohol-induced HCC) using the Kolmogorov–Smirnov test on combined networks are included in Tables S2–S5, respectively. Here, we review the top 10 significant pathways from HCC and each subtype of HCC.

Among the results of the PoTRA analysis for HCC in Table S2, we can find many pathways involved in HCC in the top 10 significant pathways, such as Cytokine-cytokine receptor interaction, cAMP signaling pathway, p53 signaling pathway, etc. Cytokine-cytokine receptor interaction exerts a vast array of immunoregulatory actions critical to cancers (*Schreiber & Walter, 2010*; *Spangler et al., 2015*). The cAMP signaling pathway regulates a number of biological processes, such as cell growth and adhesion, neuronal signaling, energy homeostasis and muscle relaxation (*Fajardo, Piazza & Tinsley, 2014*). The P53 signaling pathway is a classical oncogenic pathway as mentioned above.

Among the results for hepatitis C-induced HCC (Table S3), we find seven non-common pathways for this subtype of HCC in the top 10 pathways, such as the MAPK signaling pathway, the Calcium signaling pathway, the cGMP-PKG signaling pathway, the Adrenergic signaling in cardiomyocytes, the HIF-1 signaling pathway, the Toll-like receptor signaling pathway and the Insulin resistance. Among them, some of them are strongly related to immune system and inflammation, such as the cGMP-PKG signaling pathway, the HIF-1 signaling pathway, the Toll-like receptor signaling pathway and the Insulin resistance. These immune- and inflammation-related pathways are also specific for hepatitis C (virus infection)-induced HCC.

Among the results for hepatitis B-induced HCC (Table S4), we also find seven non-common pathways between this subtype of HCC and HCC in the top 10 pathways, such as Adrenergic signaling in cardiomyocytes, Breast cancer, Calcium signaling pathway, cGMP-PKG signaling pathway, MAPK signaling pathway, PI3K-Akt signaling pathway and Tuberculosis. Among them, there are some pathways related to immune system and inflammation, such as the cGMP-PKG signaling pathway and the PI3K-Akt signaling pathway, which are specific for hepatitis B (virus infection)-induced HCC.

Among the results for alcohol-induced HCC (Table S5), we also find four non-common pathways between this subtype of HCC and HCC in the top 10 pathways, such as the MAPK signaling pathway, the Pathways in cancer, the Calcium signaling pathway and the Progesterone-mediated oocyte maturation. Among these four pathways, the Calcium signaling pathway is shown to be specifically associated with alcohol in some studies (Gruol & Parsons, 1996; Li, Li & Guo, 2014; Bartlett et al., 2017).

As mentioned above, the combined networks, which intersect correlation networks with curated database networks, have a large loss of gene-gene interactions when compared to correlation networks. Hence, the distribution of PageRank scores of genes might be changed more from normal to cancer than that in correlation networks. Hence, more significant pathways with altered distribution of PageRank scores of genes are identified for combined networks than that for correlation networks. Although the Kolmogorov–Smirnov test identifies a large amount of significant pathways, we can use this test for combined networks to rank the pathways associated with cancer.

## DISCUSSION

We propose a PageRank-based method, Pathway of Topological Rank Analysis (PoTRA), for identifying pathways involved in cancer. PoTRA is motivated by the observation that the loss of connectivity is a common topological trait of cancer networks (Anglani et al., 2014) and the prior knowledge that a normal biological network includes a small number of hub nodes and a large number of non-hub nodes (Albert, 2005; Khanin & Wit, 2006; Zhu, Gerstein & Snyder, 2007). From normal to cancer, the process of the network losing connectivity might be the process of disrupting the structure of the network, which can result in an altered number of hub genes between normal and cancer. The PoTRA analysis is based on topological ranks of genes in biological pathways, and PoTRA detects pathways involved in cancer by testing if the number of hub genes in pathways is altered between normal and cancer.

To illustrate the method, PoTRA is applied to several TCGA hepatocellular carcinoma datasets. The results in our study are in agreement with prior knowledge of HCC from literature. We find that a high proportion of statistically significant pathways play important roles in cancer, indicating that the altered number of hub genes for these pathways might indeed be a reflection of the underlying biological causes that lead to cancer. Moreover, in the comparison between normal and each subtype of HCC, most importantly, the "Hepatitis B" pathway and several pathways associated with virus infection dramatically become significant pathways in hepatitis B-induced HCC, suggesting that PoTRA is capable of detecting pathways associated with disease subtypes. We also find several pathways associated with HCC generally and subtype specifically in hepatitis C-induced HCC and in alcohol-induced HCC.

In our approach, the correlation method is used to construct gene co-expression networks for normal and cancer, respectively. A gene co-expression network is an undirected graph, where each node represents a gene, and each edge is established if there is a significant co-expression relationship between two genes (*Stuart et al., 2003*). *Stuart et al. (2003)* found 22,163 co-expression relationships, each of which has been conserved across evolution, suggesting that the co-expressions between genes confers a selective advantage and thus these genes are functionally related. Gene co-expression networks are biologically interesting since co-expressed genes might be controlled by members of the same pathway, or the same transcriptional regulatory program or protein complex (*Weirauch, 2011*), and could be functionally related, suggesting that co-expression is common in the human genome. A gene co-expression network can be constructed by looking for pairs of genes with a similar expression pattern across samples, i.e., the transcript levels of two co-expressed genes rise and fall together across samples. As the method of network construction, we use Pearson's correlation in consideration of saving computing time, because the other methods are relatively computationally intensive. In addition, we also use combined networks by taking the intersection of KEGG curated networks and correlation networks, which increases the reliability of network construction. This approach also shows that the results using correlation networks and the results using combined networks are consistent.

Here, we construct gene networks using two ways. One approach is based on correlation networks, while the other approach is to combine (intersect) the correlation networks with pre-defined networks from pathway databases. Because the combined networks lose a large amount of gene-gene interactions from the correlation networks, the power is reduced and the PageRank scores of genes tend to more evenly distributed in combined networks than that in correlation networks. Hence, fewer significant pathways with altered number of hub genes between two phenotypes are identified for combined networks than that for correlation networks. Moreover, the distribution of topological ranks of genes might be changed more from normal to cancer than that in correlation networks. Thus, more significant pathways with altered distribution of PageRank scores of genes are identified for combined networks than that for correlation networks. Although the Kolmogorov–Smirnov test identifies a number of significant pathways for combined networks, we can use the Kolmogorov–Smirnov test to rank those pathways.

We apply the Fisher's exact test and the Kolmogorov–Smirnov test to analyze the PageRank scores for the two types of gene networks as mentioned above. The corresponding results suggest that the Fisher's exact test and the Kolmogorov–Smirnov test can identify and rank cancer-associated pathways. This suggests that the PoTRA method is robust to choice of network building approach and to statistical analysis method for identification of cancer-related pathways.

## FUTURE DIRECTIONS

The hypothesis of our study is based on the fact that the loss of connectivity is a common topological trait of cancer networks (*Anglani et al., 2014*). It is not yet well understood if this trait is a characteristic of other complex diseases. Thus, we need to be cautious about the applicability of this method to other diseases. However, this trait could be applicable to other complex diseases. Thus, although PoTRA is motivated by work on cancer, it could apply to other complex diseases as well. This area needs to be further investigated.

In this study, we apply PoTRA to pre-defined biological pathways, from the well-curated KEGG pathway database. However, the PoTRA method can also be applied to any set of genes of interest, such as functional gene subnetworks. This could be an interesting area to further explore.

In this article, we focus on the details of the PoTRA methodology. For different pathway databases, such as Reactome, Biocarta, etc., the method would apply in a similar manner, while the final results might vary slightly, depending on the data being used. This direction— changes in analysis results based on different pathway databases—would be an interesting area to investigate in the future.

## CONCLUSION

In summary, PoTRA provides a new method for detecting cancer-associated pathways. PoTRA may be used to augment existing methods and provide a richer, more systematic understanding of cancer mechanisms.

### Funding
This work was supported by Arizona State University. The funders had no role in study design, data collection and analysis, decision to publish, or preparation of the manuscript.

### Grant Disclosures
The following grant information was disclosed by the authors:
Arizona State University.

### Competing Interests
The authors declare there are no competing interests.

## Author Contributions

- Chaoxing Li conceived and designed the experiments, performed the experiments, analyzed the data, contributed reagents/materials/analysis tools, prepared figures and/or tables, authored or reviewed drafts of the paper, approved the final draft.
- Li Liu and Valentin Dinu conceived and designed the experiments, contributed reagents/materials/analysis tools, authored or reviewed drafts of the paper, approved the final draft.

## Data Availability

The following website includes code and raw data:

http://dinulab.org/tools/potra/

Code is embedded in the ''PoTRA.R'' file.

Raw data is embedded in the ''PoTRA-example-data.Rdata'' file.

## Supplemental Information

Supplemental information for this article can be found online at http://dx.doi.org/10.7717/peerj.4571#supplemental-information.

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
