# Peer review of "Pathways of topological rank analysis (PoTRA): a novel method to detect pathways involved in hepatocellular carcinoma"

_PeerJ, doi:10.7717/peerj.4571_

## Round 0.1 · original submission · Major Revisions

Dear Authors,

As you can see below the two referees have a substantial number of concerns and questions, including some of the decisions on the used datasets and validity of the gene expression variance calculation, among others.

There are also some important questions regarding the interpretation of the results in terms of biological significance.

Alfonso Valencia
PeerJ Academic Editor

Reviewer 1 ·

Basic reporting

The manuscript is clearly structured and easy to understand. An introduction to the field and literature references are provided; however, when discussing previous network analysis methods, the authors mainly present classical Gene Set Enrichment Analysis (GSEA) and approaches to identify differential co-regulation and co-expression of genes in biological networks, but previous topological network analysis methods that may provide an alternative to the authors' PoTRA approach are not mentioned (a very comprehensive review of these approaches has been provided by Mitrea et al., Frontiers Physiol., 2013, https://www.ncbi.nlm.nih.gov/pmc/articles/PMC3794382/). Some of the representative approaches for topology-based network and pathway analysis should be briefly discussed to enable the reader to make a comparison (currently, only approaches are mentioned which according to the authors "are limited by the fact that they
ignore the topology of the gene networks and sets that they investigate" (line 89), but this does not apply to all previous methods).
Apart from this, the descriptions in the text, and the figures, code and data provided are sufficient to understand and reproduce the methodology proposed by the authors.

Experimental design

The aim, scope and research question are well defined and the topic is relevant to the field. The hypothesis that the previously observed reduced network connectivity in cancer may result from a reduced number of hub genes that disrupts the scale-free structure of the network is interesting, and in principle, the proposed topological analysis is a logical approach to test the hypothesis and identify pathways with corresponding topological changes.
However, my main concern is the construction of the gene network: The authors do not use a experimentally confirmed gene regulatory or protein-protein interaction networks, but instead build correlation networks that are purely derived from Pearson correlations between gene expression values from public cancer transcriptomics datasets. High correlations between the expression values of thousands of genes in transcriptomics datasets can easily occur by chance or reflect indirect relationships rather than physical interactions - using a significance cut-off of 0.05 is not sufficient to ensure that the correlations reflect biologically meaningful functional associations in cancer or normal cells (moreover, the authors do not state whether the computed p-value has been adjusted for multiple hypothesis testing).
Importantly, the loss of hub nodes in the correlation network observed by the authors does not necessarily need to reflect the loss of a hub gene/protein in a real biological interaction network, but may instead only reflect an increased variance in the cancer gene expression data as compared to healthy samples.
Increased variance in transcriptomics datasets will nearly always result in lower Pearson correlations between genes, and therefore also in a lower number of hub genes in correlation networks, but this does not mean that a loss of hub nodes occurs in real biological interaction networks.
The hub count changes identified in genes from certain pathways using the authors' PoTRA approach may therefore reflect an increase of variance in the corresponding cancer pathways, but not necessarily a real change in hub nodes in the underlying molecular networks. To test this, the authors should also investigate whether there is a change of variance in the expression across all genes in a pathway, when comparing cancer vs. controls, and they should assess statistically, whether altered correlations between pairs of relevant genes also match with known direct regulatory or protein-protein interactions between corresponding genes/proteins, in order to check whether a hypothesized topological change in a network of real biological interactions is plausible (this cannot replace an experimental validation, but the reviewer understands that an experimental evaluation may not be feasible for the authors - therefore, additional in-silico analyses, such as the ones suggested above, should be provided and the limitations of correlation-based network analysis highlighted in the manuscript).

Validity of the findings

The overall approach is logical and described in a reproducible fashion; however, the issues associated with purely correlation-based gene networks (see Experimental design section) may also negatively affect the validity of the findings. If higher variance in the cancer gene expression data as compared to the tumor-free controls results in lower Pearson correlations between the genes in the cancer samples, then this would also explain the reduced numbers of hub nodes in the cancer correlation networks. In this case, the proposed algorithm/software would not detect new cancer-associated pathways with changes in hub nodes in a real biological network, but only changes in variance, which can also be identified by statistical variance analyses without additional network information (i.e. the same or very similar pathways would be identified by already existing tools that score changes in the variance of expression levels within pathways).
If a correlation-based network construction approach is used, then the authors need to check that the loss of hub nodes does not simply result from increased expression variance in cancer. Moreover, a more reliable correlation measure than the Pearson correlation should be considered for network construction, e.g. the biweight mid-correlation which is more robust to outliers (see Zheng et al., 2014, https://www.ncbi.nlm.nih.gov/pmc/articles/PMC4271563) or indirect and spurious correlations could partly be filtered out by using the partial correlation for network construction. If significance scores are used to define the cut-off for representing a correlation as an edge in the network, then these significance scores should be adjusted for multiple hypothesis testing (the authors use an FDR-adustment for the pathway analysis, but it is not clear whether they also apply it for the network construction).
Finally, the example applications presented by the authors are interesting, but the authors should be careful to interpret the fact that the identified pathways are cancer-associated as a validation of their approach: Generic pathways such as the discussed MAPK signalling pathway have many functions, and almost all pathways with a similarly broad involvement in many cellular processes have been implicated either directly or indirectly in cancer diseases. The fact that these pathways are identified as significant by the authors' approach does not prove that the method specifically identifies cancer-associated pathways. To assess this statistically, one possible approach could be to define a priori two lists of known cancer-associated and non-cancer associated pathways, and then evaluated statistically, whether cancer-associated pathways are over-represented among those identified by PoTRA in relation to non-cancer associated pathways.

Reviewer 2 ·

Basic reporting

The manuscript is well-written and the implemented methodology is clearly described in a step-by-step style. Some parts of the text appears redundant between separate sections, but the degree of overlap is not too much.

The references to the related works in Introduction (lines 103-118) are rather old. The authors should add more recent works on the topics of differential co-expression analysis and differential network topology analysis.

The method is based on the assumption that biological networks are scale-free, but no results to support this are shown. The scale-free property does not always hold in practice (PMID:15767579, 15284103, 16003372,16706727).

The figures are of sufficient quality, but Figure 3 remains very uninformative. The gene names are not shown at all, and most of the nodes and edges are overlapping, making it impossible to get any biological insights.

Experimental design

The topic of the paper is withing the scope of the journal. The research question is well-defined, and meaningful for cancer biology; however, there are already a plenty of methods available based on the same concepts and PageRank algorithm. It remains unclear what is the added value of this method beyond the existing ones, and why the researchers should use the proposed method? Rigorous comparative investigation against the related methods and software is required to convince the readers about its added value and performance.

Choosing only one cancer type from TCGA cannot really demonstrate the general applicability of the method. The authors should apply their method to several cancer types (and sub-types, see below), and also compare the results across various cancers and tissue origins, to investigate how cancer (sub-type)-specific or selective the findings really are.

The methodology is clearly described in a step-by-step manner, and the sources codes have been made available to allow for replication of the results. Some parts of the methods, however, lack rationale or motivation, and should be described in more detailed manner. The authors should also investigate the robustness of their results to parameter changes:

- Why KEGG databases was chosen to define pathway gene sets; have the authors evaluated how much the results change if some other pathway databases was used instead (e.g. Reactome or MSigDB)? Why the pathway interactions were not used to define the network topology, instead or in addition to the co-expression gene-gene links?

- Why Pearson's correlation was chosen to define gene co-expression interactions; have the authors evaluated how much the results change if some other metric (e.g. MI or MIC, see PMID: 23217028) was used instead? Why p<0.05 was chosen as cutoff; was this corrected p-value (FDR) or not, and what was the test (permutation or large sample assumption)?

- The Fisher's exact test requires a cut-off for testing the differences in hub genes between normal and cancer groups. Why 95th percentile was chosen, and how much would the results change if one uses some other cut-off instead? Could the Kolmogorov–Smirnov test be used alternatively, similar to GSEA, which avoids the need of introducing any cut-offs?

Validity of the findings

The current results are not very convincing as these are presented at a very general level; for instance, "Pathways in cancer" is too broad to learn any selective mechanisms for HCC or other cancer types. It remains also unclear whether the users can identify any novel, potentially cancer-specific genes or pathways using the proposed method?

The cancer sub-type-specific findings are most relevant and interesting, as most of the cancer types cannot be treated as a one entity, rather they contain various sub-types with distinct genomic or epigenetic backgrounds. The authors should follow-up these findings, and make more rigorous investigation of the relevance of pathways for various sub-types.

In the sub-type-specific analyses, the number of samples may become quite small, when going into smaller and smaller sub-types (e.g., precision medicine approach). It remains unclear how the correlation calculation deals with such small sample sizes (and what are the sample sizes), when estimating the statistical significance of co-expression interactions?

Additional comments

The method is based on an established concept that cancer processes are characterized by the loss of network connectivity (sometimes cancer processes lead also to new connections and re-wiring, which could be studied as well). However, The other assumption of scale-free property of co-expression networks is not so straightforward (e.g. one can impose the property by changing the correlation cut-off, seen PMID 17553854). Therefore, all the current results related to the scale-free property or its changes (e.g. lines 313-317) are speculative only, and should be accompanied by rigorous statistical analysis of the property in the networks constructed based on the TCGA data (see e.g. 16706727).

---

## Round 0.2 · Major Revisions

Thanks for revising the manuscript addressing some of the key critical aspects.

At this point, Reviewer 2 has additional comments for you. I believe these comments are interesting but do not affect the essential aspects of the paper. However, in light of the note below from PeerJ staff, as you will need to revise the manuscript anyway, I wanted to also give you an opportunity to respond to these comments and incorporate them as you see fit when undertaking your revision.

Reviewer 1 ·

Basic reporting

no comment

Experimental design

no comment

Validity of the findings

no comment

Additional comments

The authors have addressed my comments and their revisions have improved the manuscript. Most importantly, they acknowledge now that the cancer-associated changes in correlation networks most likely result from increased variance in the cancer gene expression data as compared to healthy samples rather than from a topological change in real molecular networks, which will help to prevent a misinterpretation of the results generated by the authors' approach.

Reviewer 2 ·

Basic reporting

The scale free property was not statistically tested in the revised paper, or did I miss it? The authors should follow the formal statistical testing procedures described in the previous papers cited in my comments, to show that their scale free assumption holds in the analyzed datasets; and if not, how this affects the operation of their approach. I do not see this as very critical assumption of the approach, but since the authors mention that many times, it should be also statistically tested.

Experimental design

If this manuscript is the first (methodological) work about the approach, I would still add the explanations of the methodological issues I asked in my original review section, rather than providing the answers in the response letter only. Please add to the Methods your answers answers to my comment: “Some parts of the methods, however, lack rationale or motivation, and should be described in more detailed manner.

The authors should also investigate the robustness of their results to parameter changes, e.g., how much their finding change if the pre-defined parameters were altered.

Validity of the findings

The current biological results are still not too convincing; rather, the current results and tables seem like lists of already known findings (which is just fine as for positive controls, but they should be described as such), or potential hypotheses that someone could follow-up (perhaps in separate papers), but making more detailed “story” for selected cases would make the results more concrete. Regarding other datasets, the authors’ response to my original comment was quite surprising:

Original comment: Choosing only one cancer type from TCGA cannot really demonstrate the general applicability of the method. The authors should apply their method to several cancer types (and sub-types, see below), and also compare the results across various cancers and tissue origins, to investigate how cancer (sub-type)-specific or selective the findings really are.

Author’s response: We are currently working on the analysis of breast cancer data and pediatric desmoplastic round cell tumor (DSRCT) cancer data using PoTRA. These data sets are still being analyzed and manuscripts are under preparation. Also, we are planning to analyze the entire TCGA data set (33 cancer types currently in GDC) and publish the results separately.

Could a few of these ongoing analyses in other cancer types be included into this work, to support the wider applicability of the method, rather than published separately? Further, the sub-type specificity of the findings is very important issue to address, like argued in my original comments, as these cancers cannot be treated as one entity. The authors should follow-up their findings, and make more rigorous investigation of the relevance of the pathways for various sub-types.

Additional comments

The authors have partly addressed my original comments. However, there still remain important concerns that were not addressed in the revised version (as described in the boxed above).

---

## Round 0.3 · accepted · Accept

Thanks for addressing the missing points. I can accept your argument for not extending the study to other cancer types at this point.

#